# Efficient Photocatalytic Hydrogen Evolution over TiO_2-X_ Mesoporous Spheres-ZnO Nanorods Heterojunction

**DOI:** 10.3390/nano10112096

**Published:** 2020-10-22

**Authors:** BingKe Zhang, Qi Li, Dongbo Wang, Jinzhong Wang, Baojiang Jiang, Shujie Jiao, DongHao Liu, Zhi Zeng, ChenChen Zhao, YaXin Liu, ZhiKun Xun, Xuan Fang, ShiYong Gao, Yong Zhang, LianCheng Zhao

**Affiliations:** 1National Key Laboratory for Precision Hot Processing of Metals, Harbin Institute of Technology, Harbin 150001, China; zhangbingke007@163.com (B.Z.); 18245019907@163.com (D.L.); 1144420106@hit.edu.cn (Z.Z.); zhaochenstu@163.com (C.Z.); lyx15545582475@163.com (Y.L.); gaoshiyong@hit.edu.cn (S.G.); yongzhang@hit.edu.cn (Y.Z.); lczhao@hit.edu.cn (L.Z.); 2Department of Optoelectronic Information Science, School of Materials Science and Engineering, Harbin Institute of Technology, Harbin 150001, China; 3Key Laboratory of Functional Inorganic Material Chemistry, School of Chemistry and Materials Science, Heilongjiang University, Ministry of Education of the People’s Republic of China, Harbin 150080, China; liqchem@sina.com; 4College of Science, Guangdong University of Petrochemical Technology, Guandu Road No. 139, Maoming 525000, China; 5Key Laboratory for Photonic and Electronic Bandgap Materials, Ministry of Education, School of Physics and Electronic Engineering, Harbin Normal University, Harbin 150025, China; 6State Key Laboratory of High Power Semiconductor Lasers, School of Science, Changchun University of Science and Technology, 7089 Wei-Xing Road, Changchun 130022, China

**Keywords:** ZnO NRs/TiO_2-X_ MSs heterojunction, photocatalytic hydrogen production, oxygen vacancies, efficiency, spectral response region

## Abstract

Photocatalytic water splitting into hydrogen is regarded as one of the key solutions to the deterioration of the global environment and energy. Due to the significantly reduced grain boundaries, ZnO nanorods facilitate a fast electron transfer through their smooth tunnels and are well suited as a photocatalyst. However, the photocatalytic hydrogen evolution performance of pristine ZnO nanorods is still low due to the high recombination rate of photogenerated electron-hole pairs and the less light absorption. Here, a novel structure about black ZnO nanorods (NRs)/TiO_2-X_ mesoporous spheres (MSs) heterojunction has been prepared and the photocatalytic hydrogen evolution performance has been explored. The photocatalytic activity test results showed that ZnO NRs/TiO_2-X_ MSs exhibited higher catalytic activity than ZnO NRs for hydrogen production. Compared to the pure ZnO NRs photoanode, the photocurrent of ZnO NRs/TiO_2-X_ MSs heterojunction photoanode could reach 0.41 mA/cm^2^ in view of the expanding spectral response region and effective inhibition of e^−^/h^+^ recombination at the same condition. Using a relatively integrated experimental investigation and mechanism analysis, we scrutinized that after being treated with NaBH_4_, TiO_2_ MSs introduce oxygen vacancies expanding the photocatalytic activity of pure TiO_2_, and improving conductivity and charge transport capabilities through coating on ZnO NRs. More importantly, the results provide a promising approach in the NRs/MSs composite structure serving as photoanodes for photocatalytic hydrogen production.

## 1. Introduction

Currently, ZnO nanostructures-based photocatalysts have attracted great attention owing to their non-toxicity, optical transparency, high photoconversion efficiency, low cost, etc. [1,2,3,4,5,6,7]. Due to the significantly reduced grain boundaries, one-dimensional (1D) nanostructures especially nanorods facilitate a fast electron transfer through their smooth tunnels [8,9,10]. Nevertheless, the photocatalytic H_2_ evolution performance of pristine ZnO nanorods is still low due to the high recombination rate of photogenerated e^−^/h^+^ pairs and the less light absorption (only in the ultraviolet region accounting for 3–5% of solar energy) [11,12,13,14,15]. To overcome the above disadvantages, the heterojunction structure based on ZnO nanorods provides a new insight for suppressing charge recombination and enhancing light absorption.

Many studies on the coupling of ZnO and TiO_2_ have shown an improved photocatalytic performance in dye degradation and hydrogen evolution [16,17,18,19,20]. Although the proper band edge position of TiO_2_ relative to ZnO can assist the effective interfacial charge transfer, the fabrication of the ZnO/TiO_2_ heterojunction cannot broaden the light absorption range, which is still too narrow to improve the efficiency of the photocatalyst. Recently, Ti^3+^ self-doped black TiO_2_ got supreme importance in the field of light harvesting, which can be formed from conventional white TiO_2_ to regulate the electronic energy band structure of TiO_2_ with the expansion of its optical response from ultraviolet light to a visible and infrared region [21,22,23,24,25,26,27,28]. It can be seen from previous reports that TiO_2_ mesoporous microspheres have an ultrahigh specific surface area and strong light scattering ability, which is achieved by combining the advantages of primary nanocrystalline composition and sub-micron structure [29,30,31,32]. However, up to now, the investigation on the photocatalytic performance of black TiO_2-X_ mesoporous microspheres/ZnO nanorods nanostructure heterojunctions is very rare.

In this research, a novel structure about black TiO_2-X_ mesoporous spheres/ZnO nanorods heterojunction has been prepared and the photocatalytic hydrogen evolution performance was explored. The light absorption region of the heterostructure covers the ultraviolet to infrared region nearly over the entire spectral range and exhibits excellent photocatalytic hydrogen production performance compared to the pure ZnO and ZnO/TiO_2_. Additionally, as confirmed from electrochemical, electronic, and spectroscopy characterizations, the significantly improved hydrogen production performance is found in the designed ZnO/TiO_2-X_ heterojunction structure, which is rooted from the potential difference generated on the ZnO-TiO_2-X_ interface and increased light absorption from the reduction of TiO_2_. The experimental results of the present work indicate a novel strategy for developing the ZnO nanostructure-based composite materials with excellent photocatalytic properties.

## 2. Materials and Methods

### 2.1. Preparation of ZnO Nanorods (NRs)

The ZnO seed layer was deposited on FTO glass (15 Ω square) by the frequency (RF) magnetron sputtering system. The FTO substrates were ultrasonically cleaned with acetone solutions, ethanol, and deionized water in an ultrasound bath for 20 min, respectively. A high-purity ZnO target (99%) with a 60.0 diameter was used for continuous sputtering at 20 W RF power for 5 min, and the working pressure was maintained at 1.0 Pa with the flow ratio of oxygen to argon 18 to 42 sccm. Then, the ZnO seed layer was annealed at 450 °C for 60 min in air.

The ZnO nanorods were grown onto the seed substrates by a simple hydrothermal method. The ZnO seed layer was immersed in a mixed solution containing 30 mL of 0.03 mmol of Zn(NO_3_)_2_·6H_2_O, 0.06 mmol of HTM in the Teflon-lined stainless steel autoclave, and then the autoclave was heated in a constant temperature electric oven at 90 °C for 6 h.

### 2.2. Preparation of ZnO Nanorods (NRs)/TiO_2-x_ Mesoporous Spheres (MSs) Composites

The TiO_2_ MSs were synthesized under hydrothermal conditions as previously reported [33]. NaBH_4_ and TiO_2_ powders were mixed in an equal ratio, and the mixture was ground for 20 min completely. Then, the mixture was heated at 300 °C for 60 min under an argon atmosphere. The mixture was repeatedly filtered and washed with water and ethanol to remove NaBH_4_. The prepared TiO_2-X_ MSs (1.0 g) was added to ethanol (80 mL) containing α-terpineol (4.34 mL) and ethyl cellulose (0.5 g) and stirred vigorously for 1 h, and then the solvent was removed with a rotary evaporator at 10 h to prepare the TiO_2-X_ paste. The spin coating solution was obtained by repeatedly stirring the slurry thirty times, and spin-coating on the ZnO nanorods at 3500 rpm for 20 s, followed by sintering at 450 °C for 30 min of preparation of the unreduced TiO_2_ and ZnO composite material followed by the same pattern.

### 2.3. Characterizations

The morphology and crystallinity of the obtained material were performed by scanning electron microscopy (SEM, Carl Zeiss, Merlin Compact, Jena, Germany). The transmission electron microscope (TEM, FEI, Hillsboro, USA) and high-resolution TEM (HRTEM, FEI, Hillsboro, USA) observations were performed with a TEI Tecnai G2 F30 (FEI, Hillsboro, USA) microscope operating at 300 KV, while energy-dispersive X-ray spectroscopy (EDS) and elemental mapping were performed on the transmission electron microscope. On the spectrometer (ESCALAB 250Xi, Thermo Scientific Escalab, Waltham, USA), the X-ray photoelectron spectroscopy (XPS) measurement was carried out. The X-ray power diffraction (XRD) analysis was characterized by an X-ray diffractometer (Empyrean, Panalytical, Malvern, UK) equipped with Cu Kα radiation (λ = 1.5418 Å). Raman spectra were obtained using a 532.8 nm excitation laser on the LabRAM HR EV0 of Horiba Jobin Yvon, Langjoux, France. The optical response of all the samples was obtained by photoluminescence (PL) measurements excited by a 325 nm He-Cd laser. At room temperature, the ultraviolet visible diffuse reflectance spectra (DRS) was recorded on the Shimadzu UV1700-visible spectrophotometer.

### 2.4. Photoelectrochemical (PEC) Measurements

PEC measurements were conducted on the Metrohm electrochemical workstation (Autolab-PGSTAT302N) in a three-electrode cell, the obtained sample, platinum foil, and Ag/AgCl electrode as the working, counter, and reference electrodes, respectively. A 1 M KOH solution was used as an electrolyte without the additive. A model LS1000-4S-AM1.5G-1000W solar simulator (Solar Light Company, Glenside, PA, USA) equipped with a metal mesh was employed to give an irradiance of 100 mW/cm^2^. Under an AM 1.5 G light source, linear sweep voltammetry (LSV) measurements were performed with a sweep rate of 1 mV/s. The photocatalytic H_2_ evolution experiments were performed in the photocatalytic system evaluation device (AuLight, Beijing, CEL-SPH2N) equipped with an online gas chromatograph (SP7800, TCD, 5 molecular sieve, Beijing Keruida Limited), and nitrogen as the carrier gas. AM 1.5 was used as the light source to trigger the photocatalytic reaction. Under stirring conditions, a certain 0.05 g catalyst was dispersed into a mixed solution composed of 100-mL methanol aqueous solution (15 vol%) and methanol as a sacrificial reagent. A certain amount of H_2_PtC_16_ solution was added dropwise to load 1 wt% Pt on the surface of the catalyst by in situ photo deposition. A gaseous sample was collected and analyzed every hour.

## 3. Results

Morphological and structural characterizations of the pure ZnO NRs and the ZnO-NRs/TiO_2_ MSs composite samples are presented in Figure 1. From the top view images of Figure 1a, it can be found that the ZnO nanorods grow upward on the FTO substrate with a high diffracted intensity and each ZnO nanorod has a typical hexagonal structure cross section. The average diameter of ZnO nanorods is about 130 nm, as shown in Figure 1b. Compared with pure ZnO nanorods, no obvious morphological changes are observed on the surface of the nanorods coated with TiO_2_ or TiO_2-X_. Obviously, TiO_2_ and TiO_2-X_ MSs are observed on the surface of the ZnO nanorods in Figure 1c–f. It is proven that the TiO_2_ and TiO_2-X_ coating solutions deposit MSs on the NRs surface after the spin-coating process. The detailed nanostructures of ZnO-TiO_2-X_ are further performed by transmission electron microscope (TEM) and high-resolution TEM (HRTEM). It can be clearly observed from Figure 2a that the TiO_2-X_ MSs adhere on the surface of the coarsened ZnO NRs and the TiO_2-X_ MSs/ZnO NRs still keep a three-dimensional morphology. The clear lattice fringes in the HRTEM image indicate that the formation of a ZnO-TiO_2-X_ heterostructure is well-crystallized. Two different types of lattice images are obtained with d spaces of 0.350 and 0.260 nm, corresponding to the (101) plane of typical anatase TiO_2_ and the (002) plane of hexagonal ZnO, respectively [34]. The hybrid photocatalyst has clear and continuous stripes, indicating that there is an intense mutual attraction between ZnO NRs and TiO_2-X_ due to suitable lattice parameters [35]. Furthermore, this also proves that the reduced TiO_2_ nanocrystals are still highly crystalline, and the Ti^3+^ introduced by the reduction does not make the crystal lattice disordered. The corresponding EDS elemental mapping images of the ZnO NRs coated with TiO_2-X_ MSs are presented in Appendix A (Appendix A) (ESI†).

Figure 3 shows the X-ray diffraction (XRD) patterns of the fabricated ZnO, ZnO/TiO_2_, ZnO/TiO_2-X_, and FTO substrate for comparison. The XRD pattern of pure ZnO nanorods displays diffraction peaks around 31.77, 34.42, 36.25, and 47.54°, which could be well indexed to the characteristic peaks (100), (002), (101), and (102) planes of the hexagonal ZnO with a wurtzite structure (JCPDS file no. 36-1451). In addition, the main peaks with a higher intensity depict the ZnO nanorods epitaxially with a very uniform orientation grown along the C-axis on the ZnO seed layer. After spin coating TiO_2_ and TiO_2-X_, the ZnO NRs also exhibit a strong wurtzite structure diffraction, but a close observation of the XRD pattern in the range of 20 to 30° reveals a broad peak centered at 25.2° (inset of Figure 3). The peaks can be assigned to the (101) of the TiO_2_ anatase phase (JCPDS 21-1272). Raman is also powerful to identify the structural changes, including determination of the presence of heterojunction surface and oxygen vacancies. Figure 4 shows the Raman spectra of the samples, where pure ZnO Raman modes that are located at 98.32 and 435.72 cm^−1^ correspond to the two typical vibration modes E_2_ (low) and E_2_ (high) of the hexagonal wurtzite structure zinc oxide crystals, respectively [36]. Obviously, four more peaks are observed in the Raman spectrum of the ZnO nanorods coated with TiO_2_. The peaks appearing at 398.94 cm^−1^, 518.31 cm^−1^ and 144.30 cm^−1^, 637.54 cm^−1^ are assigned respectively to the B_1g_ mode, a doublet of the A_1g_ and B_1g_ modes and E_g_ mode of anatase phase TiO_2_, which indicates that the ZnO nanorods are coated with TiO_2_ and TiO_2-X_ successfully [37,38,39]. This result is in agreement with those from the XRD and SEM measurements. Nevertheless, the strongest E_g_ mode area at 144.30 amplified by TiO_2_ shows a slight shift accompanied by a larger linewidth by the NaBH_4_ reduction treatment. This proves that local lattice defects related to surface oxygen vacancies will be introduced after the reduction, and a blue shift will be caused due to phonon confinement or non-stoichiometry. In addition, the broadening of the peak position is also related to the decrease of crystal quality [40,41,42]. To further perform the surface composition and chemical state of ZnO/TiO_2-X_, the XPS results are shown in Appendix A (Appendix A), ESI†. Full elemental X-ray photo electron spectroscopy proves the successful composite of ZnO with TiO_2-X_. A detailed analysis of the inner Ti 2p orbital electrons shows the Ti^4+^ 2p3/2 and 2p1/2 spin-orbit splitting peaks at 457.89 and 463.59 eV, respectively. The additional small peaks in the black TiO_2_ at 457.34 and 463.04 eV can be attributed to the 2p1/2 and 2p3/2 peaks of Ti^3+^.

Figure 5 shows the absorption spectrum of the ZnO NRs, ZnO NRs/TiO_2_ MSs, and ZnO NRs/TiO_2-X_ MSs heterojunction thin films grown on glass. All samples show a strong absorption of ultraviolet light, which can be attributed to the electronic transition from the valence band to the conduction band. It is indicated that the absorption onset of the ZnO/TiO_2-X_ is at ~400 nm, which has an obvious red shift compared with the ZnO and ZnO/TiO_2_ composite material, which may be correlated with the amount of Ti^3+^ in the shell. The optical band gaps of the samples can be determined according to the Kubelka-Munk equation: (αhν) n = K (hν − E_g_ ), where “hν” and “α” are photonic energy and the optical absorption coefficient, respectively. The bandgap of the ZnO NRs, ZnO NRs/TiO_2_ MSs, and ZnO NRs/TiO_2-X_ MSs can be assessed as ~3.21, ~3.10, ~3.02 eV, respectively by measuring the linear extrapolation to the hν-axis (Appendix A (Appendix A), ESI†) [43]. More importantly, the absorption of the sample combined with the TiO_2-X_ mesoporous sphere is markedly increased at a 400–600 nm region, compared with pure ZnO the enhancement of sunlight harvesting can be attributed to the lattice disorder introduced in the formation of black TiO_2_ (the generation of mid gap energy levels within the band gap) [44].

To judge the quality of ZnO crystals and perform the mechanism of photoinduced charge carriers and interfacial charge transfer over the ZnO/TiO_2-X_ heterostructures, the room temperature PL spectra of the three synthesized samples are shown in Figure 6. The pure ZnO nanorods show a strong and sharp ultraviolet luminescence peak at ~372 nm, originating from the near band gap excitonic emission, and the other is located at ~738 nm, which is attributed to the second-order diffraction peak emitted at the ultraviolet band [45]. No obvious defect emission is seen in the yellow-green wavelength band of visible light, indicating that the ZnO nanorods prepared in this way have good crystallinity and fewer surface defects [46,47]. After coating with TiO_2_, the emission intensity is significantly reduced, which indicated that the composite materials reduce the recombination probability of photogenerated e^−^/h^+^ pair. It is worth noting that the black TiO_2-X_ mesoporous spheres/ZnO nanorods heterojunction has the lowest electron-hole recombination rate. Therefore, the unique heterostructure photocatalyst of the ZnO/TiO_2-X_ has a positive effect on the effective separation of photogenerated carriers during the photocatalytic reactions charge carriers. This considering the fact that the number of oxygen vacancies in titanium dioxide is too low to be disclosed, it can be easily explained that the significantly defect emission peak by oxygen vacancies in the PL was not observed. It is worth noting that the emission peak of ZnO nanorods shifted to the right after coating with TiO_2-X_. This behavior may be due to the quantum confinement stark effect [48,49,50]. Therefore, the type II band structure of ZnO/TiO_2-X_ promotes the excited carriers, which confines the electron-hole recombination, and results in the emission intensity decreases. From the electron dynamics point of view, the observed lifetime shortenings obviously indicate an improvement in the charge separation for the ZnO/TiO_2-X_, leading to an increase in photocatalytic performance (as below).

To perform the effect of ZnO/TiO_2-X_ heterojunction on photoelectrochemical properties, the linear sweep voltammetry (LSV) behavior of these samples is recorded under a simulated AM 1.5 sunlight illumination (Figure 7a). The photocurrent starting potential of ZnO/TiO_2-X_ is −0.637 vs. Ag/AgCl, which shows a slight shift compared with −0.618 vs. Ag/AgCl of ZnO/TiO_2_ and the fastest growth of anodic current at a higher applied voltage. It is probably related to the best electron transportation and improved charge transfer compared to the pure ZnO electrode and ZnO/TiO_2_. In addition, a set of linear scans are collected in the dark, as shown in Appendix A (Appendix A), ESI†. All three photoelectrodes show a very low dark current. The transient photocurrent response can also predict the photocatalytic activity of the electrode. The transient photocurrent response analysis is employed under illumination with several 20 s light on/off cycles at 0 vs. Ag/AgCl in Figure 7b. Transient photocurrent response experiments are explored to prove the enhancement in separation and transport efficiency of charge carriers in ZnO/TiO_2-X_ nanostructures. It can be observed that the photocurrent signal can be repeatedly switched from the “on” state to the “off” state by periodically turning the light source on and off for all electrodes, which shows that all electrodes have a good stability. It is worth noting that after coating with TiO_2-x_ the photocurrent has been significantly enhanced compared with pure ZnO nanorods. Simultaneously, the photocurrent induced by the simulated solar light irradiation of ZnO/TiO_2-X_ (0.41 mA/cm^2^) at 0 vs. Ag/AgCl, is about eight times larger than that of ZnO (0.05 mA/cm^2^). The pure ZnO exhibited a much lower photocurrent than ZnO/TiO_2-X_, which may be attributed to the high recombination rate of photo-induced charges and weak light absorption. Under light, the narrow band gap of TiO_2-X_ can capture low-energy photons. The narrow band gap of TiO_2-X_ can increase the light absorption rate and the internal electrostatic field in the ZnO/TiO_2-X_ junction causes the recombination rate of photogenerated e^−^/h^+^ to decrease.

Using methanol as the electron donor and Pt as the co-catalyst, the photocatalytic hydrogen evolution experiment is carried out under AM 1.5. Figure 8 shows the results of hydrogen evolution experiments of ZnO and ZnO/TiO_2-X_ materials. After 5 h of xenon lamp irradiation, ZnO/TiO_2-X_ nanocomposites exhibited significant photocatalytic activities towards H_2_ production performance, and its hydrogen evolution amount reaches 243.4 μmol·g^−1^ better than the previous reports [51,52]. The addition of TiO_2-X_ has a significant effect on the photocatalytic activity of ZnO, indicating that the TiO_2-X_ hollow structures could promote more effective photons absorption and prolong the lifetime of the photo-induced electrons to a certain extent.

As revealed in the above experimental results of ZnO/TiO_2-X_, the plausible photocatalytic mechanism is proposed for the improved hydrogen production activity, as shown in Figure 9. Since the valence bands (VB) positions of TiO_2-X_ are lower than that of ZnO before TiO_2-X_ is contacted with ZnO, and the conduction band (CB) positions of pure ZnO are higher than that of TiO_2-X_. For this type-Ⅱ structure in our work, under the simulated solar light irradiation, electrons accumulate in the CB of TiO_2-X_ and the holes are transferred from TiO_2_ to ZnO (in VB), respectively. Moreover, black TiO_2-X_ nanostructures with oxygen vacancies enhance light absorption under full sunlight. On the other hand, oxygen vacancies caused by the reduction can be considered as electron donors of TiO_2-X_, which can further effectively separate and transport light-excited electron-hole pairs. This proper band alignment can be applied to separate the active photogenerated electrons and holes, thus increasing the photocatalytic H_2_ rate.

## 4. Conclusions

In summary, a novel structure on black TiO_2-X_ mesoporous spheres/ZnO nanorods heterojunction has been prepared and the photocatalytic hydrogen evolution performance has been explored. The morphology, composition, and crystallinity test results confirmed the formation of a clear hybrid structure on the FTO substrate. The properties of the ZnO NRs/TiO_2-X_ MSs hetero-junction and the photocatalytic performance have been carefully performed through various experimental methods. The photocurrent and hydrogen production performance of ZnO NRs/TiO_2-X_ MSs have been significantly enhanced compared with pure ZnO NRs. The solar-driven hydrogen evolution rate of ZnO NRs/TiO_2-X_ MSs is several times higher than that of ZnO. All these suggest that these novel ZnO NRs/TiO_2-X_ MSs are prospective, next-generation photocatalytic materials of low-cost, large area, and energy-efficiency for practical applications.

## Figures and Tables

**Figure 1 nanomaterials-10-02096-f001:**
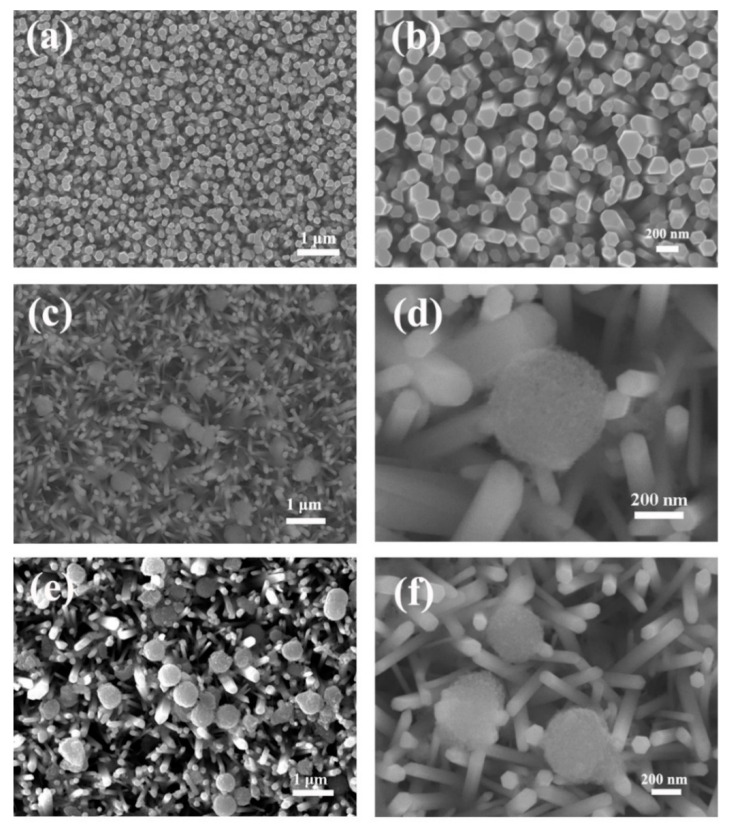
Typical top view SEM images of pure ZnO (**a**,**b**), ZnO/TiO_2_ (**c**,**d**), ZnO/TiO_2-X_ (**e**,**f**) samples on the FTO substrate.

**Figure 2 nanomaterials-10-02096-f002:**
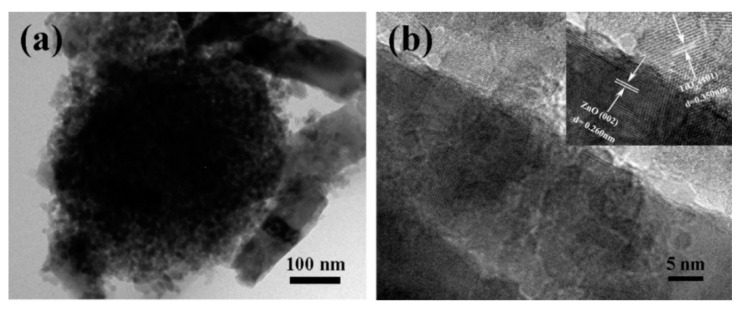
Structural characterizations of ZnO NRs/TiO_2-X_ MSs: (**a**) TEM image of ZnO NRs/TiO_2-X_ MSs; (**b**) HRTEM image of ZnO NRs/TiO_2-X_ MSs.

**Figure 3 nanomaterials-10-02096-f003:**
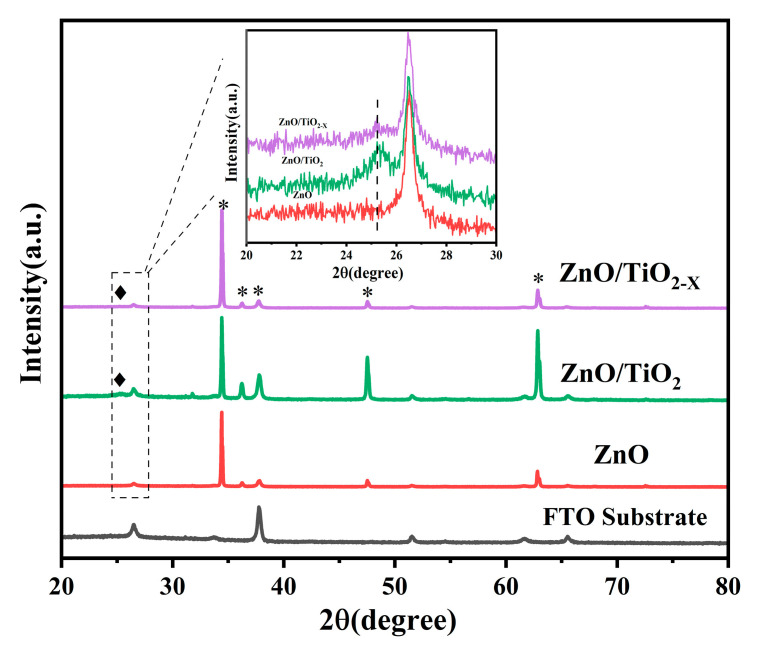
XRD patterns of ZnO, ZnO/TiO_2_, ZnO/TiO_2-X_, and FTO substrate samples (* represents the peak position of the hexagonal ZnO with a wurtzite structure and ♦ the peak position of TiO_2_ anatase phase).

**Figure 4 nanomaterials-10-02096-f004:**
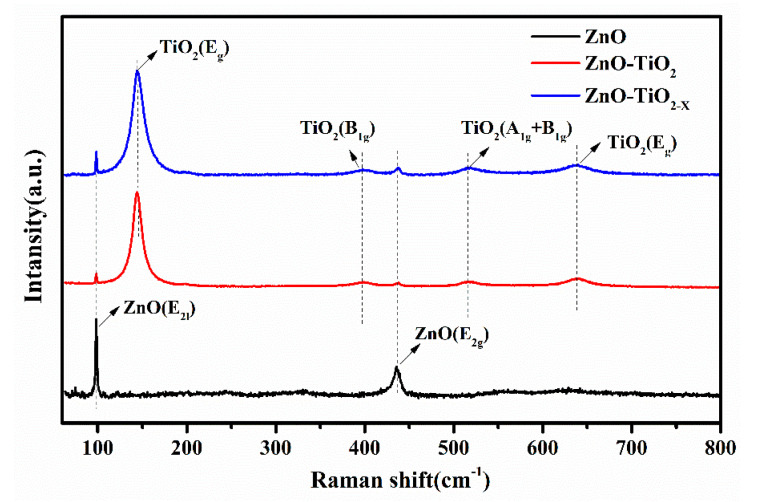
Raman spectra of TiO_2_ NRs, ZnO/TiO_2_, ZnO/TiO_2-X_.

**Figure 5 nanomaterials-10-02096-f005:**
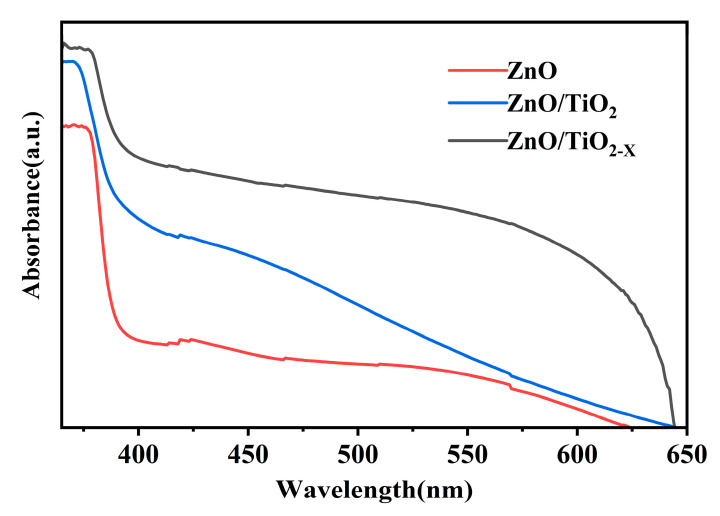
UV-Vis diffuse reflectance spectra of the pure ZnO, ZnO/TiO_2_, ZnO/TiO_2-X_ samples.

**Figure 6 nanomaterials-10-02096-f006:**
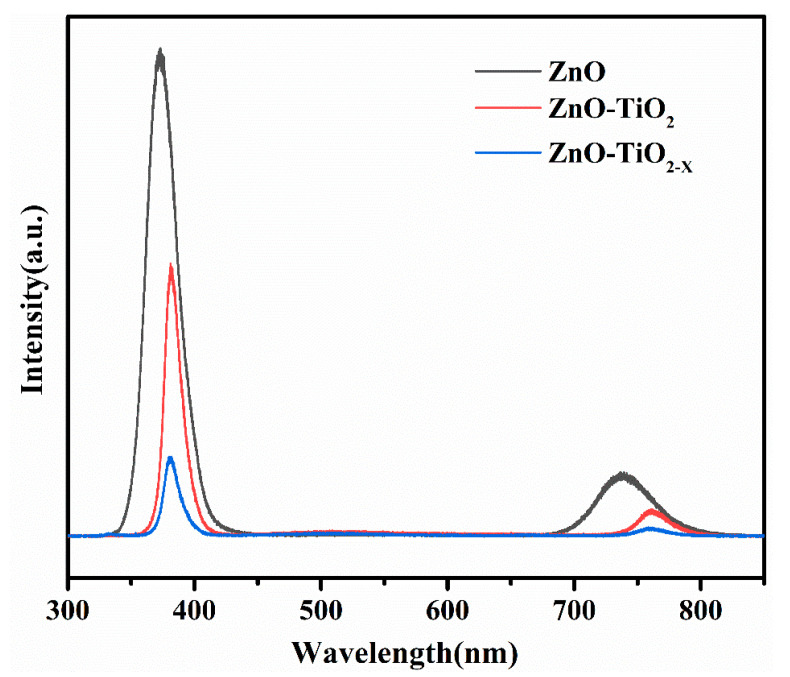
PL spectra of different photoanodes recorded with the excitation wavelength of 325 nm.

**Figure 7 nanomaterials-10-02096-f007:**
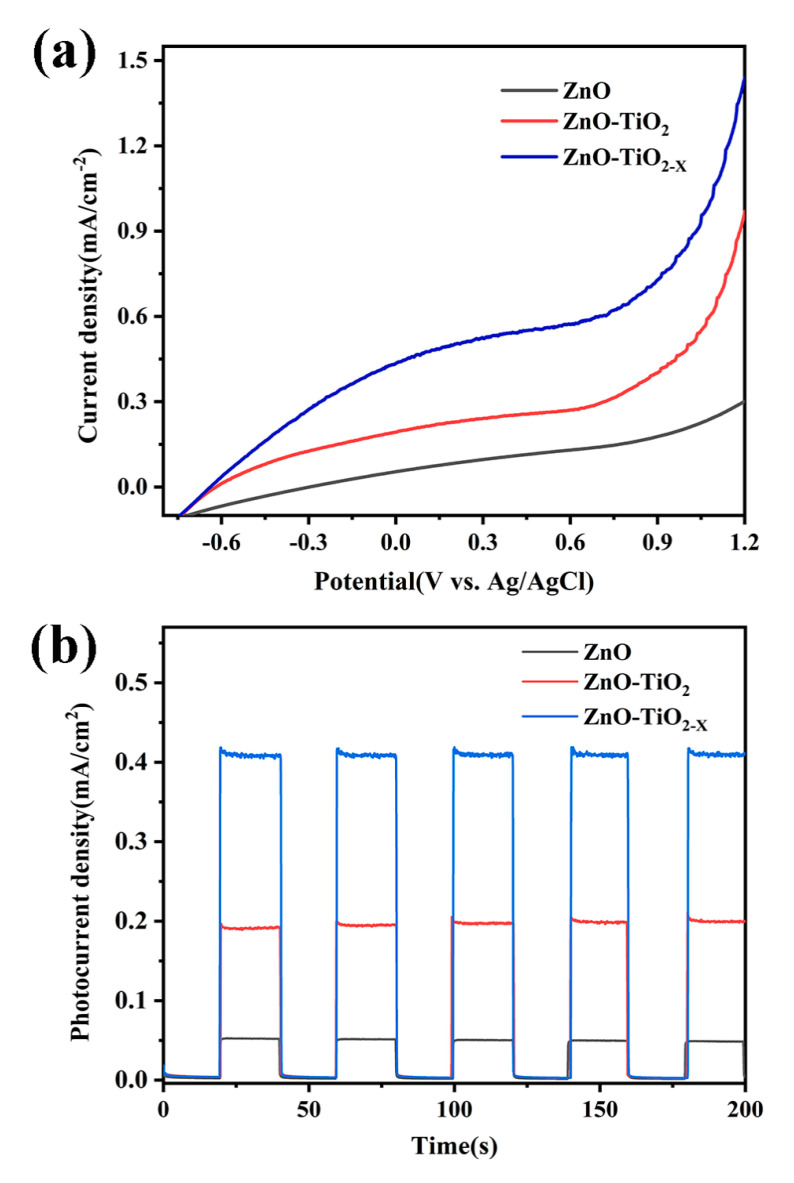
(**a**) Linear sweep voltammetry (LSV) characteristics and (**b**) photocurrent responses of pure ZnO, ZnO/TiO_2_, ZnO/TiO_2-X_ homojunction with on/off radiation in a 1 M KOH solution under AM 1.5.

**Figure 8 nanomaterials-10-02096-f008:**
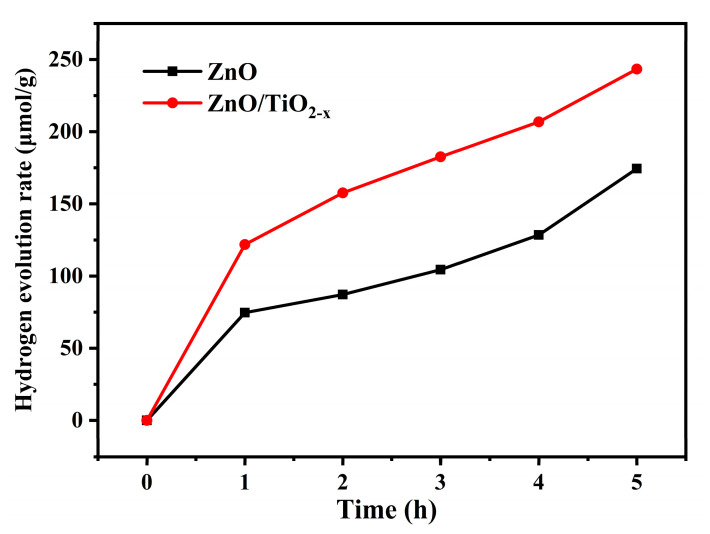
Temporal H_2_ evolution of the pure ZnO and ZnO/TiO_2-X_ under AM 1.5.

**Figure 9 nanomaterials-10-02096-f009:**
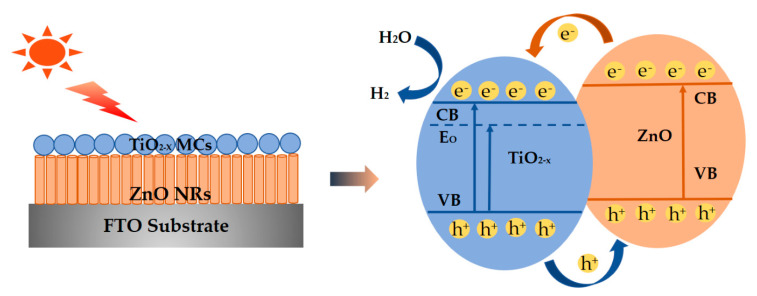
Mechanism model for the enhanced property of ZnO/TiO_2-X_ composite photocatalysts for the photocatalytic H_2_ evolution reaction.

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
