# Peer review of "Efficient Photocatalytic Hydrogen Evolution over TiO2-X Mesoporous Spheres-ZnO Nanorods Heterojunction"

_nanomaterials, 2020, doi:10.3390/nano10112096_

Round 1

Reviewer 1 Report

This is a study related to the development of ZnO/TiO2-x heterojunctions. It is a brief work with a straightforward characterization, leading to improved performance of the combined materials both as photoanode and as photocatalyst for H2 generation.

The synthesis methods are simple and the structures obtained show the formation of ZnO nanorods, decorated with mesoporous TiO2-x spheres, which overperfom the pristine ZnO and, in the case of the PEC measurements, the ZnO/TiO2 heterojunctions.

Despite the obvious improvement, the work has several weaknesses: 
1. As the authors state, this kind of systems have been widely studied. Is the novelty of the work, the combination with black titania? The synthesis method? The values obtained? There are few recent works reporting these heterojunctions.

2. There intrinsic differences between the photoelectrochemical and the photocatalytic tests. While evaluated as photoanodes, there is a significant improvement versus pristine ZnO nanorods, there is a doubt about how the performance would be compared to TiO2 or TiO2-x. Would there be a difference in the onset potentials?

3. Regarding the photocatalytic, it would be necessary to study the materials after the "in situ" Pt photodeposition. It is possible that the deposition of Pt does not occur in the same manner on the single ZnO and on the the ZnO/TiO2-x system, which later on would also affect the H2 formation. Moreover, carrying this process in this "in situ" approach, can mask both situations, as methanol photoreforming can also lead to H2 formation. Certainly, it would be better to firstly deposit Pt (in a comparable way), and later on, to evaluate the H2 generation either as overall water photosplitting catalysts (if possible) or in a photoreforming test.

4. Besides these comments, the reviewer would suggest:
- Include dark curves in the LSV of Figure 7.
- Please correct the photocurrent value in page 8, line 229 (it's 0.05 mA/cm2). 
- Which is the origin of the absorption below 650 nm, especially in the ZnO sample?
- How were the photocatalysts tests carried out? With the materials supported on glass-FTO?
- Please indicate which is the "spin coating solution".  

Author Response

Dear editor and reviewers:
Thank you very much for your careful review and constructive suggestions with regard to our manuscript “Efficient Photocatalaytic Hydrogen Evolution over TiO2-X mesoporous Spheres-ZnO nanorods Heterjunction” (ID: nanomaterials-953980). Those comments are helpful for authors to revise and improve our paper. We have studied comments carefully and tried our best to revise and improve the manuscript and made modifications and corrections in the manuscript according to the referees′ good comments, which we hope meet their approval. Revised portion is marked in blue in the paper. The main corrections in the paper and the responds to the reviewer’s comments are as flowing.

Responds to the reviewer’s comments:
Reviewer #1:

Comments and Suggestions for Authors

This is a study related to the development of ZnO/TiO2-x heterojunctions. It is a brief work with a straightforward characterization, leading to improved performance of the combined materials both as photoanode and as photocatalyst for H2 generation.

The synthesis methods are simple and the structures obtained show the formation of ZnO nanorods, decorated with mesoporous TiO2-x spheres, which overperfom the pristine ZnO and, in the case of the PEC measurements, the ZnO/TiO2 heterojunctions.

Despite the obvious improvement, the work has several weaknesses: 

Comment 1. As the authors state, this kind of systems have been widely studied. Is the novelty of the work, the combination with black titania? The synthesis method? The values obtained? There are few recent works reporting these heterojunctions.

Response: Thank you for the suggestion. The innovation of this paper is the combination of ZnO nanorods and TiO2-X mesoporous spherical heterojunction. At present, no combination of this ZnO and TiO2-X related morphology heterojunction has been found. TiO2-x is obtained by reduction with sodium borohydride. TiO2-x maintains the anatase structure and morphology of white titanium dioxide after reduction. The content of Ti3+ has been verified by XPS test, and its relative content has been proved semi-quantitatively by peak area. Relevant research has been added to the supporting information.

Comment 2. There intrinsic differences between the photoelectrochemical and the photocatalytic tests. While evaluated as photoanodes, there is a significant improvement versus pristine ZnO nanorods, there is a doubt about how the performance would be compared to TiO2 or TiO2-X. Would there be a difference in the onset potentials?

Response: Thank you for the suggestion. Through the formation of heterojunctions between ZnO nanorods and TiO2 and TiO2-X, which can indicate the difference in performance between the TiO2 or TiO2-X. The LSV research shows that the onset potential of the TiO2 base is -0.618V, and the onset potential of the TiO2-X base is -0.637V.

Comment 3. Regarding the photocatalytic, it would be necessary to study the materials after the "in situ" Pt photodeposition. It is possible that the deposition of Pt does not occur in the same manner on the single ZnO and on the the ZnO/TiO2-x system, which later on would also affect the H2 formation. Moreover, carrying this process in this "in situ" approach, can mask both situations, as methanol photoreforming can also lead to H2 formation. Certainly, it would be better to firstly deposit Pt (in a comparable way), and later on, to evaluate the H2 generation either as overall water photosplitting catalysts (if possible) or in a photoreforming test.

Response: Thank you for the suggestion. This opinion is very meaningful to us, because the deadline is tight, we will study this aspect in depth in future experiments. Thank you very much for your advice. In follow-up research, we will conduct a lot of experiments to study related issues.

  • Lian, Z.C.; Wang, W.C.; Li, G.S.; Tian, F.H.; Schanze, K.S.; Li, H.X. Pt-Enhanced Mesoporous Ti3+/TiO2 with Rapid Bulk to Surface Electron Transfer for Photocatalytic Hydrogen Evolution. ACS APPL MATER INTER,2017, 9,20,16960-16967.
  • Zou, X.X.; Liu, J.K.; Su, J.; Zuo, F.; Chen, J.S.; Feng, P.Y. Facile Synthesis of Thermal-and Photostable Titania with Paramagnetic Oxygen Vacancies for Visible-Light Photocatalysis.2013, 19,8, 2866-2873.
  • Hu, W.Y.; Zhou, W.; Zhang, K.F.; Zhang, X.C.; Wang, L.; Jiang, B.J.; Tian, G.H.; Zhao, D.Y .; Fu, H.G. Facile strategy for controllable synthesis of stable mesoporous black TiO2 hollow spheres with efficient solar-driven photocatalytic hydrogen evolution J. Mater. Chem. A,2016, 4,19,7495-7502.
  • Tan, H.Q.; Zhao, Z.; Niu, M.; Mao, C.Y.; Cao, D.P.; Cheng, D.J.; Feng, P.Y.; Sun, Z.C. A facile and versatile method for preparation of colored TiO2with enhanced solar-driven photocatalytic activity, Nanoscale,2014, 6, 17,10216-10223.
  • Song, H.; Li, C.X.; Lou, Z.R.; Ye, Z.Z.; Zhu, L.P. Effective Formation of Oxygen Vacancies in Black TiO2 Nanostructures with Efficient Solar-Driven Water Splitting. ACS Sustain. Chem. Eng.2017, 5,10,8982-8987.
  • Sui, Y.L.; Liu, S.BA.; Li, T.F.; Liu, Q.X.; Jiang, T.; Guo, Y.F.; Luo, J.L. Atomically dispersed Pt on specific TiO2 facets for photocatalytic H2 evolution. J Catal. 2017, 353,250-255.
  • Carabineiro, S.A.C.; Machado, B. F.; Drazic, G.; Bacsa, R. R.; Serp, P.; Figueiredo, J. L.; Faria, J. L. Photodeposition of Au and Pt on ZnO and TiO2.2013,175, 629-633.
  • Pawinrat, P.; Mekasuwandumrong, O.; Panpranot, J. Synthesis of Au-ZnO and Pt-ZnO nanocomposites by one-step flame spray pyrolysis and its application for photocatalytic degradation of dyes.2009, 10,10,1380-1385.

According to the following related studies, it is shown that platinum can be deposited on ZnO and TiO2/TiO2-X in the same form by photo-initiated in-situ deposition.

Comment 4. Besides these comments, the reviewer would suggest:

- Include dark curves in the LSV of Figure 7.

Response: Thank you for the suggestion, the dark current test chart has been attached to the support information.

Fig. 5. Linear sweep voltammogram under dark conditions

Comment 5. Please correct the photocurrent value in page 8, line 229 (it's 0.05 mA/cm2).

Response: Thank you for underlining this deficiency, the error message has been changed.

Comment 6. Which is the origin of the absorption below 650 nm, especially in the ZnO sample?

Response: The absorption before 400nm comes from the absorption of the main band gap, and the absorption below 650 nm comes from the N diffusion from air into the ZnO layer at the interface.

  1. S. G. Zhang, X. W. Zhang, Z. G. Yin, J. X. Wang, J. J. Dong, Z. G. Wang, S. Qu, B. Cui, A. M. Wowchak, A. M. Dabiran and P. P. Chow, Improvement of electroluminescent performance of n-ZnO/AlN/p-GaN light-emitting diodes by optimizing the AlN barrier layer. J. Appl. Phys., 2011, 109, 093708
  2. Ü. Özgür, Ya. I. Alivov, C. Liu, A. Teke, M. A. Reshchikov, S. Doğan, V. Avrutin, S.-J. Cho, and H. Morkoç A comprehensive review of ZnO materials and devices, J. Appl. Phys. 98, 041301 (2005).

Comment 7. How were the photocatalysts tests carried out? With the materials supported on glass-FTO?

Response: Thank you, ZnO and its composite materials are grown on FTO substrates for direct photoelectrochemical testing. The three sample materials were scraped off from FTO for hydrogen production test.

Comment 8. Please indicate which is the "spin coating solution".

Response: Thank you for your very useful advice, the detailed preparation process of ZnO/TiO2 samples has been modified. Add TiO2/black TiO2 powder, ethyl cellulose and terpineol to the ethanol solution, make a slurry by rotary steaming, and make a spin coating liquid by ultrasonic stirring.

Reviewer 2 Report

The manuscript reports on the production of a new ZnO/TiO2-x heterojunction and its use as photocatalyst for the H2 production. The manuscript is well arranged scientifically and contains much new information in the field of photocatalytic water splitting and for these reasons, I consider the paper suitable for publication, but before this, some minor revisions should be done. I suggest to improve the entire manuscript taking into account the following indications: 

Entire manuscript:

  • The work plan is correct and the manuscript is quite easy to follow, but, anyway, I suggest a linguistic revision and a proper check for some grammar errors (plurals and verbs). Some sentences need to be rewritten (for example lines: 71-75, 91-94, 102-103, 166-169, 200-203). The use of acronyms should be revised, sometimes some acronyms are used without a proper introduction in the text (for example MC, PEC, HER NW correspond to what? please correct where needed). Sometimes within the sentences some words start with capital letter, please check also this aspect.

Experimental section:

  • Details about the production of ZnO/TiO2 sample should be provided.
  • Why textural properties of the materials have not been evaluated?
  • To quantify Ti3+ amount in ZnO/TiO2-x heterojunction photoemission spectroscopy should be considered.
  • The possible reuse of the photocatalyst for water splitting tests has been proved?
  • Please specify the meaning of the acronym PEC

Results and Discussion section:

  • Lines 139-141: the reported datum is in agreement with the following recent paper: J. Phys. Chem. C 2020, 124, 3564−3576. Please consider it as reference for the production of self-doped TiO2.
  • Line 206: please add a reference for “quantum confinement Stark effect”
  • Line 222: I suggest to change the verb “to investigate” with the verb “to perform”
  • H2 production: a comparison with literature data on other ZnO heterojunctions should be provided to underline the good performances of the proposed material.
  • Line 265: change the word "Discussion" with the word "Conclusions"
  • Just a curiosity: which are the possible poisoning agents for the proposed catalysts?

Author Response

Dear editor and reviewers:
Thank you very much for your careful review and constructive suggestions with regard to our manuscript “Efficient Photocatalaytic Hydrogen Evolution over TiO2-X mesoporous Spheres-ZnO nanorods Heterjunction” (ID: nanomaterials-953980). Those comments are helpful for authors to revise and improve our paper. We have studied comments carefully and tried our best to revise and improve the manuscript and made modifications and corrections in the manuscript according to the referees′ good comments, which we hope meet their approval. Revised portion is marked in blue in the paper. The main corrections in the paper and the responds to the reviewer’s comments are as flowing.

Responds to the reviewer’s comments:

Reviewer #2:

The manuscript reports on the production of a new ZnO/TiO2-x heterojunction and its use as photocatalyst for the H2 production. The manuscript is well arranged scientifically and contains much new information in the field of photocatalytic water splitting and for these reasons, I consider the paper suitable for publication, but before this, some minor revisions should be done. I suggest to improve the entire manuscript taking into account the following indications: 

Comment 1. Entire manuscript:

The work plan is correct and the manuscript is quite easy to follow, but, anyway, I suggest a linguistic revision and a proper check for some grammar errors (plurals and verbs). Some sentences need to be rewritten (for example lines: 71-75, 91-94, 102-103, 166-169, 200-203). The use of acronyms should be revised, sometimes some acronyms are used without a proper introduction in the text (for example MC, PEC, HER NW correspond to what? please correct where needed). Sometimes within the sentences some words start with capital letter, please check also this aspect.

Response: Thank you for underlining these deficiencies, related errors have been corrected.

Comment 2. Experimental section:

Details about the production of ZnO/TiO2 sample should be provided.

Response: Thanks, the detailed preparation process of ZnO/TiO2 samples has been modified.

Preparation of ZnO nanorods(NRs). ZnO seed layer were deposited on FTO glass (15 Ω square) by frequency (RF) magnetron sputtering system. The FTO substrates were ultrasonically cleaned with acetone solutions, ethanol and deionized water in an ultrasound bath for 20 min, respectively. A high-purity ZnO target (99%) with a 60.0 diameter was used for continuous sputtering at 20 W RF power for 5 minutes, and the working pressure was maintained at 1.0 Pa with the flow ratio of oxygen to argon is 18 sccm to 42 sccm. Then, the ZnO seed layer was annealed at 450℃ for 60 min in air.

Comment 3. To quantify Ti3+ amount in ZnO/TiO2-x heterojunction photoemission spectroscopy should be considered.

Response:  Thank you for your suggestion. XPS test was performed on the black titanium dioxide material to prove the existence of trivalent titanium, and the relative content of trivalent titanium was proved semi-quantitatively by the peak area. Related research has been added to the supporting information.

Fig. 3. XPS survey spectrum of the ZnO/TiO2-X. Inset: exact XPS analysis of the inner Ti 2p orbital electrons(peak area Ti3+/Ti4+ =0.0758)

Comment 4. The possible reuse of the photocatalyst for water splitting tests has been proved?

Response:  Thank you for your suggestion. In this experiment,the material can produce hydrogen stably within five hours and has good stability. Due to the limitation of experimental equipment, we haven't done the experiment yet. We are also actively looking for test equipment to complement this experiment.

Comment 5. Please specify the meaning of the acronym PEC

Response:  Thank you for your suggestion, PEC means photoelectrochemical test. The article has been modified.

Photoelectrochemical (PEC) measurements. PEC measurements were conducted on the Metrohm electrochemical workstation (Autolab-PGSTAT302N) in a three-electrode cell, the obtained sample, platinum foil and Ag/AgCl electrode as the working, counter and reference electrodes, respectively. A 1M KOH solution was used as electrolyte without additive. A model LS1000-4S-AM1.5G-1000W solar simulator (Solar Light Company, USA)  equipped with a metal mesh was employed to give an irradiance of 100 mW/cm2.Under AM 1.5G light source, linear sweep voltammetry (LSV) measurements were performed with a sweep rate of 1 mV/s. The photocatalytic H2 evolution experiments were performed in the photocatalytic system evaluation device (AuLight, Beijing,CEL-SPH2N) equipped with online gas chromatograph (SP7800, TCD, 5 molecular sieve, Beijing Keruida Limited), and nitrogen as the carrier gas. AM1.5 was used as the light source to trigger the photocatalytic reaction. Under stirring conditions, a certain 0.05 g catalyst was dispersed into a mixed solution composed of 100-mL methanol aqueous solution (15 vol.%) and methanol as sacrificial reagent. A certain amount of H2PtC16 solution was added dropwise to load 1 wt.% Pt on the surface of the catalyst by in-situ photodeposition. A gaseous sample was collected and analyzed every hour.

Comment 6. Results and Discussion section:

Lines 139-141: the reported datum is in agreement with the following recent paper: J. Phys. Chem. C 2020, 124, 3564−3576. Please consider it as reference for the production of self-doped TiO2.

Response:  Thanks for the suggestion, relevant documents have been added.

Comment 7. Line 206: please add a reference for “quantum confinement Stark effect”

Response:  Thanks for the suggestion, relevant documents have been added.

Comment 8. Line 222: I suggest to change the verb “to investigate” with the verb “to perform”

Response:  Thanks for the suggestion, the verb has been revised.

Comment 9. H2 production: a comparison with literature data on other ZnO heterojunctions should be provided to underline the good performances of the proposed material.

Response: Thanks for your suggestion, the relevant comparison has been added.

Comment 10. Line 265: change the word "Discussion" with the word "Conclusions"

Response:  Thanks for the suggestion, the verb has been revised.

Comment 11. Just a curiosity: which are the possible poisoning agents for the proposed catalysts?

Response: Thank you for your question, in this experiment, the main materials ZnO and TiO2 have no toxicity, and nearly no poisoning agents were used in the experiment.

Round 2

Reviewer 1 Report

The authors have considered the comments of the reviewers and have added more experimental details in the present version. Additionally, in the response, they have clarified some points and have added two results: XPS analysis and dark LSV results. Both of them have been also mentioned in the manuscript. However, the Supporting Information has not been updated, so it must be before acceptance. 

In general, the present version has been improved.

Author Response

Dear editor and reviewers:
Thank you very much for your careful review and constructive suggestions with regard to our manuscript “Efficient Photocatalaytic Hydrogen Evolution over TiO2-X mesoporous Spheres-ZnO nanorods Heterjunction” (ID: nanomaterials-953980). Those comments are helpful for authors to revise and improve our paper. We have studied comments carefully and tried our best to revise and improve the manuscript and made modifications and corrections in the manuscript according to the referees′ good comments, which we hope meet their approval. The main corrections in the paper and the responds to the reviewer’s comments are as flowing.

Responds to the reviewer’s comments:
Reviewer #1:

Comments and Suggestions for Authors

The authors have considered the comments of the reviewers and have added more experimental details in the present version. Additionally, in the response, they have clarified some points and have added two results: XPS analysis and dark LSV results. Both of them have been also mentioned in the manuscript. However, the Supporting Information has not been updated, so it must be before acceptance.

In general, the present version has been improved: 

Response: Thank you for underlining this deficiency. We changed the support information and re-uploaded it.

Please feel free to contact us with any questions and we are looking forward to your consideration.

Best regards.

Yours sincerely,

Dongbo Wang
